# TEST-TIME GRAPH REBIRTH: SERVING GNN GENERALIZATION UNDER DISTRIBUTION SHIFTS

## ABSTRACT

Distribution shifts between training and test graphs typically lead to the decreased performance of well-trained graph neural network (GNN) models, negatively affecting their ability to generalize in real-world applications. Although there have been advances in addressing graph distribution shifts through various model architectures and training strategies, implementing existing solutions in practical GNN deployment and serving at test time can be challenging, as they often necessitate significant modifications or retraining of the GNNs. To address such challenges, in this work, we propose a novel method, *i.e.*, **T**est-**T**ime **G**raph **REB**irth, dubbed **TT-GREB**, to effectively generalize the well-trained GNN models to the test-time graphs under distribution shifts by directly manipulating the test graph data. Concretely, we develop an overall framework designed by two principles, corresponding to two sub-modules: (1) *prototype extractor* for re-extracting the environment-invariant features of the test-time graph; and (2) *environment refiner* for re-fining the environment-varying features to explore the potential shifts. Furthermore, we propose a dual test-time graph contrastive learning objective with an effective iterative optimization strategy to obtain optimal prototype components and environmental components of the test graph. By reassembling these two components, we obtain a newly reborn test graph, which is better suited for generalization on the well-trained GNN model with shifts in graph distribution. Extensive experiments on real-world graphs under diverse test-time distribution shifts verify the effectiveness of the proposed method, showcasing its superior ability to manipulate test-time graphs for better GNN generalization ability.

## 1 INTRODUCTION

Recent advances in graph neural networks (GNNs) have achieved great success with promising learning abilities for various graph structural data in numerous real-world applications (Zhang et al., 2022; Zheng et al., 2022a;b; 2023c;a; Jin et al., 2022; Zheng et al., 2022c). Well-designed GNN models are ultimately intended for practical deployment and serving on various graph learning tasks (Zheng et al., 2023b; Liu et al., 2023b; Yu et al., 2023). However, these expertly trained GNNs generally experience significant performance degradation due to the *graph distribution shift* issue between the training and the test graphs (Wu et al., 2022b; Liu et al., 2023a; Chen et al., 2023b; Yu et al., 2023). The main reason behind such distribution mismatch lies in the underlying environmental variations, with time-related attribute changes, agnostic corruptions, and inconsistent graph data collection procedures (Sui et al., 2023; Chen et al., 2023b; Jin et al., 2023). These factors would lead to considerable differences in node contexts, graph structures, and the overall scale of graphs during the test-time stage.

To overcome the model generalization challenge caused by graph distribution shifts, there is a growing focus on research into learning with distribution shifts on graphs, from the *model-centric* perspective (Wu et al., 2022b; You et al., 2023; Chen et al., 2023c; Wu et al., 2020). Typically, these existing methods incorporate invariant representation learning into GNN development through customizing model architectures and training strategies (Xu et al., 2019; Zhu et al., 2021; Liu et al., 2022; Wu et al., 2022a). However, in real-world GNN deployment and serving, it may not be always practical to re-design GNN model architectures or re-train well-trained GNN models by test-time fine-tuning. This would be more challenging when these well-trained GNNs are continuously in service online, as accessing and modifying their parameters becomes more difficult. Given

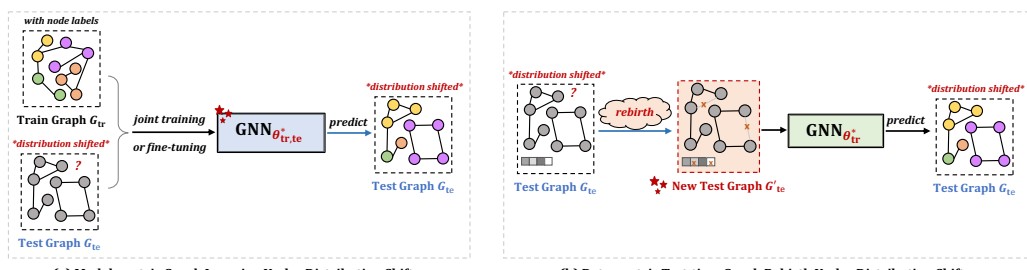

Figure 1: Comparison between the (a) model-centric graph learning methods v.s. our proposed (b) data-centric test-time graph rebirth method under graph distribution shifts.

all these circumstances including *complex training-test graph distribution shifts* and *inaccessible online-serving GNN model fine-tuning*, an intriguing problem emerges:

> **Question:** Is it possible to give the rebirth of the test-time graph data from the data-centric perspective, to serve various well-trained GNN models for better generalization under distribution shifts?

In this work, we provide a feasible graph data-centric solution to answer this question by enhancing the test-time graph data quality, without accessing the training graph data or modifying the well-trained GNNs, as shown in Fig. 1. Specifically, the original test graph would undergo a rebirth process to emerge as a new test graph, which is then fed into the well-trained GNN model for direct inference without retraining or fine-tuning. In the test-time graph rebirth process, we identify two essential components that decomposed a graph under distribution shifts: (a) *the environment-invariant component* retains consistent informative features (such as node class label prototypes) across various training and test graph distributions; (b) *the environment-varying component* dictates the extent of shifts in graph data distribution between training and test.

In light of this, we propose a novel test-time graph rebirth method for serving good GNN generalization under graph distribution shifts with two principles: (1) re-extracting the environment-invariant features of the test-time graph for identifying the predictive pattern of node class label prototypes with a **prototype extractor**, and (2) re-fining the environment-varying features to explore the potential shifts in the training-test distribution with a **environment refiner**, leading to the alignment of the training-test environment latent space with improved generalization capabilities for online GNN deployment. Furthermore, due to the unknown test-time graph node labels, we propose a dual test-time graph contrastive learning objective with self-supervision signals, along with an effective iterative optimization strategy to obtain expressive prototype features and environmental features of the test graph. In this way, we preserve prototype features to determine the node class label's predictive patterns, meanwhile, we adjust the environmental features of the original test graph to match those of the training graph. Then, these components are reassembled into a newly reborn test graph, which better suits the well-trained GNNs' generalization under distribution shifts.

In summary, the contributions of this work are as follows:

- **Graph Data-centric Paradigm.** We introduce a novel graph data-centric paradigm, Test-Time Graph REBirth (TT-GREB), designed to enhance the generalization ability of well-trained GNNs to real-world test graphs experiencing distribution shifts at test time.

- **Innovative Solution.** We develop a comprehensive framework with two core components: a prototype extractor that identifies invariant features within test-time graphs, and an environment refiner that adjusts varying features to align the latent spaces of training and test environments. These components are effectively optimized through a dual test-time graph contrastive learning objective and an iterative optimization strategy.

- **Comprehensive Evaluation.** We evaluate the proposed method on real-world test graphs under diverse graph distribution shifts. Extensive experimental results reveal consistently strong generalization ability on various well-trained GNN models, providing compelling evidence for the efficacy of the proposed method.

## 2 RELATED WORK

**Test-time Adaption.** Our work is related to test-time adaption methods, which aim to enable dynamic tuning of the pre-trained model to generalize and adapt well to the test samples (Sun et al., 2020; Wang et al., 2020; Chen et al., 2023a; Liang et al., 2020; Jang et al., 2022). The pioneering work is Test-Time Training (TTT) (Sun et al., 2020), which re-trains model weights at test time on a single test image sample through a self-supervised learning objective with an auxiliary task. Moreover, TENT (Wang et al., 2020) proposes a fully test-time adaption problem that only accesses model parameters and the test data. Considering most existing methods are model-centric and serve the computer vision domain, which might not fit GNN models and graph data, GTRANS (Jin et al., 2023) first proposes to adapt test graph data without accessing the training procedure and GNN architectures. However, GTRANS employs a fully parameterized matrix to represent modified test-time graph node features and engages in graph structure learning within the dynamic node representation learning process. For one thing, the fully parameterized approach to node feature learning merely adds to the original node features, which narrows the scope for learning updated test graphs. For another thing, GTRANS uses a binary-space projected gradient descent method, limiting the flexibility in handling diverse graph structures. In this work, we conduct the parameterized decomposition of the test graph to give it a rebirth by re-extracting the invariant features and re-fining test-time varying features of test graphs. This allows for more adaptable modifications to test graphs with an expended optimization space of our proposed TT-GREB.

**Graph Learning Under Distribution Shifts.** Our work is also relevant to the research topic of graph learning under distribution shifts, whose goal is to develop a GNN model for better generalization ability on test graphs under graph distribution shifts (Wu et al., 2020; Chen et al., 2023b; Sui et al., 2023; Guo et al., 2023; Chen et al., 2022a; Wang et al., 2022). Typically, UDAGCN (Wu et al., 2020) conducts unsupervised graph domain adaption with a domain adversarial method to learn domain-invariant embeddings across the source domain and the target domain. AIA (Sui et al., 2023) proposes graph data augmentation with effective GNN learning to handle the covariate shift on graphs for the graph classification task. Different from these graph model-centric methods, in this work, we mainly focus on modifying the test graph data with self-supervision signals to deal with the distribution-shifted test graph learning problem. Therefore, the critical distinction between our work and existing methods is that our work is primarily concerned with a graph data-centric learning paradigm to directly manipulate test graph data for serving better the GNN generalization ability under distribution shifts.

## 3 TEST-TIME GRAPH REBIRTH (TT-GREB)

**Preliminary.** Given a training graph $G_{\text{tr}} = (\mathbf{X}_{\text{tr}}, \mathbf{A}_{\text{tr}}, \mathbf{Y}_{\text{tr}}) \sim P_{\text{tr}}$ with $N$ nodes and $C$-classes of node labels, where $\mathbf{X}_{\text{tr}} \in \mathbb{R}^{N \times d}$ is the $d$-dimension nodes feature matrix indicating node attribute semantics, $\mathbf{A}_{\text{tr}} \in \mathbb{R}^{N \times N}$ is the adjacency matrix indicating whether nodes are connected or not by edges with $\mathbf{A}_{\text{tr}}^{i,j} = \{0, 1\} \in \mathbb{R}$ for $i$-th and $j$-th nodes, $\mathbf{Y}_{\text{tr}} \in \mathbb{R}^{N \times C}$ denotes the node labels, and $P_{\text{tr}}$ is the training graph distribution.

**Training Stage:** A GNN model is trained on $G_{\text{tr}}$ according to the following objective function for node classification:

$$\boldsymbol{\theta}_{\text{tr}}^* = \min_{\boldsymbol{\theta}_{\text{tr}}} \mathcal{L}_{\text{cls}} \left( \hat{\mathbf{Y}}_{\text{tr}}, \mathbf{Y}_{\text{tr}} \right), \text{ where}$$
$$\mathbf{Z}_{\text{tr}}, \hat{\mathbf{Y}}_{\text{tr}} = \text{GNN}_{\boldsymbol{\theta}_{\text{tr}}}(\mathbf{X}_{\text{tr}}, \mathbf{A}_{\text{tr}}). \tag{1}$$

The parameters of GNN trained on $G_{\text{tr}}$ is denoted by $\boldsymbol{\theta}_{\text{tr}}$, $\mathbf{Z}_{\text{tr}} \in \mathbb{R}^{N \times d_1}$ is the output node embedding of graph $G_{\text{tr}}$ from $\text{GNN}_{\boldsymbol{\theta}_{\text{tr}}}$, and $\hat{\mathbf{Y}}_{\text{tr}} \in \mathbb{R}^{N \times C}$ denotes the output node labels predicted by the trained $\text{GNN}_{\boldsymbol{\theta}_{\text{tr}}}$. By optimizing the node classification loss function $\mathcal{L}_{\text{cls}}$ (*e.g.*, cross-entropy loss) between GNN predictions $\hat{\mathbf{Y}}_{\text{tr}}$ and ground-truth node labels $\mathbf{Y}_{\text{tr}}$, the GNN model that is well-trained on $G_{\text{tr}}$ can be denotes as $\text{GNN}_{\boldsymbol{\theta}_{\text{tr}}^*}$ with optimal weight parameters $\boldsymbol{\theta}_{\text{tr}}^*$. Note that once we obtain the optimal $\text{GNN}_{\boldsymbol{\theta}_{\text{tr}}^*}$ that has been well-trained on $G_{\text{tr}}$, the GNN model would be **fixed** and $G_{\text{tr}}$ would not be accessible during test time.

**Test-time Inference:** For practical GNN deployment and serving, given a real-world test graph $G_{\text{te}} = (\mathbf{X}_{\text{te}}, \mathbf{A}_{\text{te}}) \sim P_{\text{te}}$ including $M$ nodes with its feature matrix $\mathbf{X}_{\text{te}} \in \mathbb{R}^{M \times d}$ and its adjacency

matrix $\mathbf{A}_{\text{te}} \in \mathbb{R}^{M \times M}$, we assume that there are potential distribution shifts between $G_{\text{tr}}$ and $G_{\text{te}}$, which mainly lies in node contexts, graph structures, and scales as $P_{\text{tr}} \neq P_{\text{te}}$, but the label space keeps consistent under the covariate shift, *i.e.*, all nodes in $G_{\text{te}}$ are constrained in the same $C$-classes as $G_{\text{tr}}$ as $\{\mathbf{Y}_{\text{tr}}, \mathbf{Y}_{\text{te}}\} \in \mathcal{Y} = \{1, \cdots, C\}$. Generally, the well-trained model $\text{GNN}_{\boldsymbol{\theta}_{\text{tr}}^*}$ would be directly used for inferring on the test graph as $\hat{\mathbf{Y}}_{\text{te}} = \text{GNN}_{\boldsymbol{\theta}_{\text{tr}}^*}(\mathbf{X}_{\text{te}}, \mathbf{A}_{\text{te}})$. However, due to the latent graph distribution shifts at the test time, the optimal parameters $\boldsymbol{\theta}_{\text{tr}}^*$ learned on the training graph would not be ideal for inference on the test-time graph. This can result in less accurate node classification on the test graph and does harm to the GNN's generalization ability.

Compared with existing model-centric methods working on learning optimal GNN parameter $\boldsymbol{\theta}_{\{\text{tr,te}\}}^*$ on the joint distribution of the training and test graphs, which requires GNN architecture re-designed and fine-tuned, in this work, we pay attention to a data-centric solution through modifying the test graph $G_{\text{te}}$ under distribution shifts by test-time graph rebirth, without re-designing and fine-tuning the well-trained $\text{GNN}_{\boldsymbol{\theta}_{\text{tr}}^*}$, and without accessing the training graph $G_{\text{tr}}$.

### 3.1 PROBLEM FORMULATIONS

Through the length of graph structural data generation hypothesis in existing studies (Gui et al., 2022; Ye et al., 2022; Wu et al., 2021; Sui et al., 2023; Chen et al., 2022b; 2023b), a graph can be generated through a mapping $f_{\text{gen}} : \{\mathcal{C}, \mathcal{S}\} \rightarrow \mathcal{G}$, where $\mathcal{C} \subseteq \mathbb{R}^{n_c}$ and $\mathcal{S} \subseteq \mathbb{R}^{n_s}$ are the latent variables denotes the environment-invariant part and the environment-varying parts for generating the graph $G \in \mathcal{G} = \cup_{N=1}^{\infty} \{0,1\}^N \times \mathbb{R}^{N \times d}$, where $n_c$ and $n_s$ denote the dimensions of latent variables, respectively. Inspired by such structural causal model (SCM) (Chen et al., 2022b; 2023b) for graph generation progress, we have the following proposition to comprehend test-time graph distribution shifts and the test-time graph discrepancy from the training graph.

**Proposition 1** *Given the training graph $G_{tr} \sim P_{tr}$, there exist the environment-invariant part $G_{tr}^{Inv}$ and the environment-varying part $G_{tr}^{Env}$ components in this graph, denoting as $G_{tr} = \{G_{tr}^{Inv}, G_{tr}^{Env}\}$, likewise for the the test graph $G_{te} \sim P_{te}$ with $G_{te} = \{G_{te}^{Inv}, G_{te}^{Env}\}$. Then, the GNN model $\text{GNN}_{\boldsymbol{\theta}_{tr}^*}$ that has been well trained on the $G_{tr}$ would keep good generalization on the test graph when*

$$G_{tr}^{Inv} = G_{te}^{Inv} \sim Q^{Inv}, \quad dist\left(G_{tr}^{Env}, G_{te}^{Env}\right) < \epsilon,$$
$$\text{where } G_{tr}^{Env} \neq G_{te}^{Env}, G_{tr}^{Env} \sim Q_{tr}^{Env}, G_{te}^{Env} \sim Q_{te}^{Env}. \tag{2}$$

In this proposition, the function $dist(\cdot)$ quantifies the discrepancy between the components of the training graph and the test graph that vary with the environment. Additionally, $Q^{\text{Inv}}$ represents the distribution of latent variables that remain constant across different environments, and is expected to be identical in both the training and test graphs. Furthermore, the environmental variables of $G_{\text{tr}}^{\text{Env}}$ and $G_{\text{te}}^{\text{Env}}$ are presumed to adhere to distinct, environment-specific distributions, denoted as $Q_{\text{tr}}^{\text{Env}}$ and $Q_{\text{te}}^{\text{Env}}$, respectively.

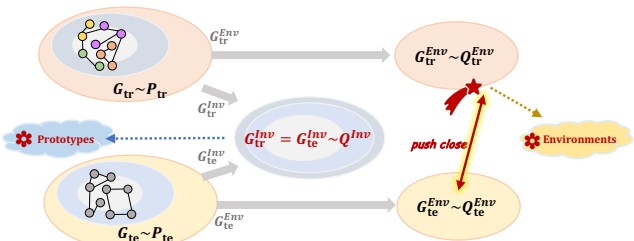

Figure 2: Illustration of distribution shifts in test-time graphs v.s. training graphs.

As shown in Fig. 2, we elaborate on the distribution shifts in the test-time graph and the training graph, from the view of latent variable decomposition according to Proposition 1. It shows two fundamental insights and principles for test-time graph rebirth:

1. Re-extracting the environment-invariant features of the test-time graph, which shares the same informative characteristics with the training graph, to assure that they can reflect the predictive pattern of node class labels, denoting as class-related **prototype** features.

2. Re-fining the environment-varying features, which are primarily attributed to possible shifts in the training-test distribution, referred to **environment** features. Essentially, a well-trained GNN model is supposed to perform expressively on the test graph, when the test distribution closely matches

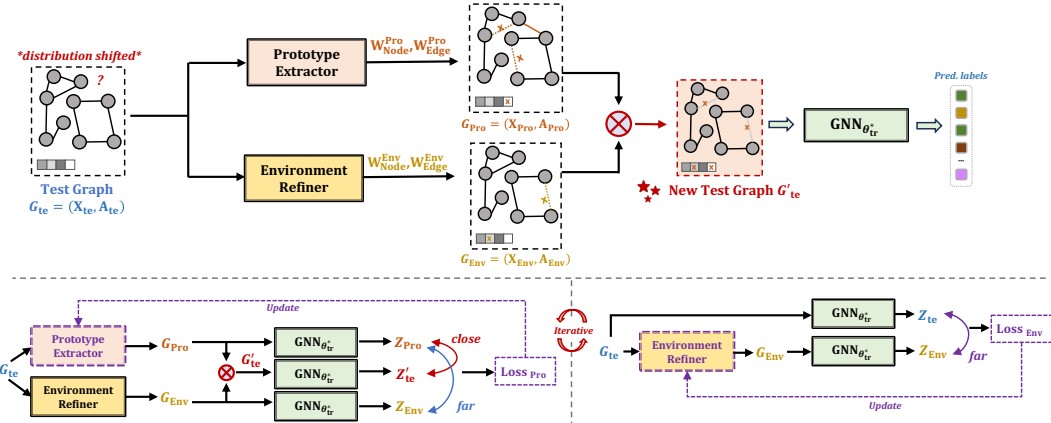

Figure 3: The overall framework of the proposed test-time graph rebirth (TT-GREB) method.

the training graph's distribution. When we push the test-time environment feature distributions more closely with those of the training graph, the well-trained GNN is likely to exhibit improved generalization capabilities under distribution shifts.

According to such two principles, if we (1) keep prototype features and (2) align the environment features on the test graph, then, make a re-composition, we could transform the original test graph to a new test graph, this process can be defined as the problem of test-time graph rebirth:

**Definition 3.1 (Test-time Graph Rebirth)** *Given the test graph $G_{te} = (\mathbf{X}_{te}, \mathbf{A}_{te})$ and the well-trained GNN model $GNN_{\theta^*_{tr}}$, test-time graph rebirth aims to learn following mapping functions: $f_{Pro} : G_{te} \to G_{te}^{Inv}$ and $f_{Env} : G_{te} \to G_{te}^{Env}$, with the re-composition function $g = f_{Pro} \circ f_{Env}$, the rebirth test graph can be denoted as*

$$G'_{te} = (\mathbf{X}'_{te}, \mathbf{A}'_{te}) = g(f_{Pro}(G_{te}), f_{Env}(G_{te})). \tag{3}$$

*In this way, the rebirth test graph would be fed into the well-trained GNN model that has been deployed online in practice for making inference $\hat{\mathbf{Y}}'_{te} = GNN_{\theta_{tr}}(\mathbf{X}'_{te}, \mathbf{A}'_{te})$, where $\hat{\mathbf{Y}}'_{te}$ is expected to be more closely aligned with the actual ground-truth node labels of the test graph compared to the initial predictions $\hat{\mathbf{Y}}_{te}$.*

### 3.2 METHODOLOGY

According to the two principles, in this work, we propose a novel method, named TT-GREB, to address the test-time graph rebirth problem for serving good GNN generalization under distribution shifts at the test time.

The overall framework is illustrated in Fig. 3. Concretely, the proposed TT-GREB consists of two components: (1) a prototype extractor identifies features that remain unchanged in different environments, mainly determined by node class labels, where these features can be consistent in both training and test graphs and reflect the predictive pattern of GNN models; and (2) an environment refiner adjusts the environment-varying features of the test-time graph, to match the latent distribution of the training graph's environment. This alignment ensures that the GNN, which is well-trained on the training graph, demonstrates strong generalization capability on the rebirth test graph. More details of the modular design of our proposed TT-GREB are presented below.

#### 3.2.1 MODULAR DESIGN.

Given a test-time graph $G_{te} = (\mathbf{X}_{te}, \mathbf{A}_{te})$ with $M$ nodes with $d$ dimension node attribute features, the prototype extractor $f_{\phi_p}^{Pro}(\cdot)$ and the environment refiner $f_{\phi_e}^{Env}(\cdot)$, parameterized by $\phi_p$ and $\phi_e$, respectively, take it as the input simultaneously. For ease of reference, we denote them as $f_{\phi_p}(\cdot)$ and $f_{\phi_e}(\cdot)$, by omitting the superscripts in subsequent mentions.

Concretely, these two sub-modules share the same structure, *i.e.*, two full-connected layers, $\text{FC}^k_{\text{node}}(\cdot)$ and $\text{FC}^k_{\text{edge}}(\cdot)$ to generate the soft and dense node attribute reweight matrix $\mathbf{W}^k_{\text{node}} \in \mathbb{R}^{d \times d}$, and the edge reweight matrix $\mathbf{W}^k_{\text{edge}} \in \mathbb{R}^{M \times M}$, where $k = \{\text{Pro}, \text{Env}\}$ for indicating the layers in the prototype extractor and the environment refiner, respectively. In this way, we have:

$$\mathbf{W}^k_{\text{node}} = \sigma\left(\text{FC}^k_{\text{node}}(\mathbf{X}_{\text{te}})\right),$$
$$\mathbf{W}^k_{\text{edge}}(i,j) = \sigma\left(\text{FC}^k_{\text{edge}}\left(\left[\mathbf{x}^i_{\text{te}}, \mathbf{x}^j_{\text{te}}\right]\right)\right), \tag{4}$$

where $\sigma(\cdot)$ is the sigmoid function that constrains both the node attribute reweight matrix and the edge reweight matrix to $[0,1]$, and $\mathbf{x}^i_{\text{te}} = \mathbf{X}_{\text{te}}[i,:]$ denotes the $i$-th row's feature representation for node $i$, so as for node $j$ with $\mathbf{x}^j_{\text{te}} = \mathbf{X}_{\text{te}}[j,:]$, attributing the value in the location $(i,j)$ for $\mathbf{W}^k_{\text{edge}}$.

Then, we could obtain the graph $G_{\text{Pro}} = (\mathbf{X}_{\text{Pro}}, \mathbf{A}_{\text{Pro}})$ that reflects the node class label prototypes and the graph $G_{\text{Env}} = (\mathbf{X}_{\text{Env}}, \mathbf{A}_{\text{Env}})$ that adjusts the test-time graph characteristics that vary with environments under distribution shifts, as:

$$G_{\text{Pro}} = f_{\boldsymbol{\phi}_p}(G_{\text{te}}) = \left(\mathbf{A}_{\text{te}} \odot \mathbf{W}^{\text{Pro}}_{\text{edge}}, \mathbf{X}_{\text{te}} \odot \mathbf{W}^{\text{Pro}}_{\text{node}}\right),$$
$$G_{\text{Env}} = f_{\boldsymbol{\phi}_e}(G_{\text{te}}) = \left(\mathbf{A}_{\text{te}} \odot \mathbf{W}^{\text{Env}}_{\text{edge}}, \mathbf{X}_{\text{te}} \odot \mathbf{W}^{\text{Env}}_{\text{node}}\right), \tag{5}$$

where $\odot$ is the broadcasted element-wise product. After these, we make a re-composition with the prototype graph $G_{\text{Pro}}$ and the environment graph $G_{\text{Env}}$ components to build a new test-time graph $G'_{\text{te}} = (\mathbf{X}'_{\text{te}}, \mathbf{A}'_{\text{te}})$ through:

$$G'_{\text{te}} = g(G_{\text{te}}) = \left(\mathbf{A}_{\text{te}} \odot \mathbf{W}^{\text{Comp}}_{\text{edge}}, \mathbf{X}_{\text{te}} \odot \mathbf{W}^{\text{Comp}}_{\text{node}}\right), \text{ where}$$
$$\mathbf{W}^{\text{Comp}}_{\text{edge}} = \left(\mathbf{1}^{\text{edge}} - \mathbf{W}^{\text{Pro}}_{\text{edge}}\right) \odot \mathbf{W}^{\text{Env}}_{\text{edge}} + \mathbf{W}^{\text{Pro}}_{\text{edge}}, \text{ and} \tag{6}$$
$$\mathbf{W}^{\text{Comp}}_{\text{node}} = \left(\mathbf{1}^{\text{node}} - \mathbf{W}^{\text{Pro}}_{\text{node}}\right) \odot \mathbf{W}^{\text{Env}}_{\text{node}} + \mathbf{W}^{\text{Pro}}_{\text{node}}.$$

In this process, $\mathbf{1}^{\text{node}}$ and $\mathbf{1}^{\text{edge}}$ denote the all-one matrices. Given $\mathbf{W}^{\text{Pro}}_{\text{node}}$ represents the class-prototype node attribute reweight matrix, $\left(\mathbf{1}^{\text{node}} - \mathbf{W}^{\text{Pro}}_{\text{node}}\right)$ can be viewed as a plain and straightforward proportion of the environment-sensitive node attributes. Then, $\left(\mathbf{1}^{\text{node}} - \mathbf{W}^{\text{Pro}}_{\text{node}}\right) \odot \mathbf{W}^{\text{Env}}_{\text{node}}$ re-composes node attribute reweight matrix by explicitly imposing the environment refinement $\mathbf{W}^{\text{Env}}_{\text{node}}$ on the environment-sensitive proportion. And then, $+\mathbf{W}^{\text{Pro}}_{\text{node}}$ makes sure to preserve the environment consistent proportion. The edge reweight matrix composition would follow the same rule. The rationale for such a re-composition schema is based on the understanding that the interplay between node class label prototypes and environment features is typically more complex than a basic additive combination, such as $\left(\mathbf{W}^{\text{Env}}_{\text{node}} + \mathbf{W}^{\text{Pro}}_{\text{node}}\right)$. This complexity becomes particularly evident under the distribution shifts encountered during test time.

By this re-composition schema, we jointly keep prototype features and align the environment features on the test graph, leading to a transformation from the original test graph to a newly reborn test graph. In this way, the new test-time rebirth graph can make effective predictions with good generalization ability on the well-trained GNN model with graph data distribution shifts.

### 3.2.2 OPTIMIZATION OBJECTIVE.

During the test time, to improve the well-trained GNN model generalization under graph distribution shifts, the significant challenge faced with graph data-centric transformation through test-time graph rebirth is the scarcity of test ground-truth labels. Consequently, this makes it more challenging to conduct supervised learning by minimizing the cross entropy loss, which is the most readily and straightforward solution. Therefore, with (1) the insufficient node class labels of the test graph, (2) the inaccessible training graph for online GNN deployment, and (3) the unknown graph distribution shifts, it is imperative to develop an effective self-supervised learning objective along with an appropriate optimization strategy.

In this work, we propose a dual test-time graph contrastive learning objective with an effective iterative optimization strategy. For one thing, we use self-supervise signals from the well-trained GNN's output node representations to guide the learning of prototype extractor with the graph

contrastive learning loss $\mathcal{L}_{\text{Pro}}$, following the parameter-free principle (Jin et al., 2023). Through the lens of the general graph contrastive learning scheme, the core idea is to maximize the similarity between two consistent views of the same graph, and to minimize the similarity when the views are not in agreement. For another thing, we perform a decomposition of environment-varying features on the test graph, by ensuring the environmental discrepancy under the graph distribution shifts during the test time. Considering the inaccessible training graph, we encourage the discrepancy between the environmental characteristics of the reborn test-time graph and the original test-time graph, leading to the graph environment refinement loss $\mathcal{L}_{\text{Env}}$ to optimize the proposed environment refiner. As illustrated in the lower section of Fig.3, these two optimization objectives are iteratively refined using gradient descent until they reach convergence. More implement details of the dual learning objective of our proposed TT-GREB are presented as follows.

Given the obtained prototype graph $G_{\text{Pro}}$, the environment graph $G_{\text{Env}}$, and the new rebirth graph $G'_{\text{te}}$, we fed them into the well-trained GNN model simultaneously, leading to the node representations with $\mathbf{Z}_{\text{Pro}} = \text{GNN}_{\boldsymbol{\theta}^*_{\text{tr}}}(G_{\text{Pro}})$, $\mathbf{Z}_{\text{Env}} = \text{GNN}_{\boldsymbol{\theta}^*_{\text{tr}}}(G_{\text{Env}})$, and $\mathbf{Z}_{\text{te}'} = \text{GNN}_{\boldsymbol{\theta}^*_{\text{tr}}}(G'_{\text{te}})$. Then, the learning objective of structural prototype feature extraction for test-time graph rebirth can be:

$$\min_{\boldsymbol{\phi}_p} \mathcal{L}_{\text{Pro}} = \sum_{i=1}^{M} \left( 1 - \frac{(\mathbf{z}_i^{\text{te}'})^\top \mathbf{z}_i^{\text{Pro}}}{\|(\mathbf{z}_i^{\text{te}'})^\top\| \|\mathbf{z}_i^{\text{Pro}}\|} \right) - \sum_{i=1}^{N} \left( 1 - \frac{(\mathbf{z}_i^{\text{Env}})^\top \mathbf{z}_i^{\text{Pro}}}{\|(\mathbf{z}_i^{\text{Env}})\| \|\mathbf{z}_i^{\text{Pro}}\|} \right) \quad (7)$$
$$+ \alpha \left[ Reg\left(\mathbf{W}_{\text{node}}^{\text{Pro}}\right) + Reg\left(\mathbf{W}_{\text{edge}}^{\text{Pro}}\right) \right].$$

Furthermore, the learning objective of environment refinement can be written as:

$$\max_{\boldsymbol{\phi}_e} \mathcal{L}_{\text{Env}} = dist(G_{\text{te}}, G_{\text{Env}}) = \|\mathbf{Z}_{\text{te}} - \mathbf{Z}_{\text{Env}}\|_2^2$$
$$- \beta \left[ Reg\left(\mathbf{W}_{\text{node}}^{\text{Env}}\right) + Reg\left(\mathbf{W}_{\text{edge}}^{\text{Env}}\right) \right], \quad (8)$$

where $Reg(\mathbf{W}^{\clubsuit}) = |\frac{\sum \mathbf{W}^{\clubsuit}}{\sum(1-\mathbf{W}^{\clubsuit})} - \lambda_s|$ is the regularization term with $s = \{1, 2, 3, 4\}$ and superscript $^\clubsuit$ corresponding to $\mathbf{W}_{\text{node}}^{\text{Pro}}$, $\mathbf{W}_{\text{edge}}^{\text{Pro}}$, $\mathbf{W}_{\text{node}}^{\text{Env}}$, and $\mathbf{W}_{\text{edge}}^{\text{Env}}$, respectively. Here, $\lambda_s$ acts as a hyper-parameter ranging $[0, 1]$, while $\alpha$ and $\beta$ balance the loss functions between the primary optimization objectives and the regularization terms. These regularization terms are designed to keep the average ratio of the number of reweighted node features or edges close to $\lambda_s$, thereby stabilizing the training process and avoiding trivial solutions.

## 4 EXPERIMENT

In this section, we verify the effectiveness of the proposed TT-GREB in terms of the GNN generalization ability on test-time graphs under distribution shifts. Concretely, we aim to answer the following questions to demonstrate the effectiveness of the proposed TT-GREB: **Q1:** How does the proposed TT-GREB perform on the well-trained GNNs for node classification task under various graph distribution shifts at test time? **Q2:** How does the proposed TT-GREB perform when conducting an ablation study regarding the sub-module components and the learning strategy? **Q3:** How sensitive are the hyper-parameter $\lambda$ for the proposed TT-GREB? **Q4:** How does the proposed TT-GREB perform in terms of running time efficiency and visualization?

### 4.1 EXPERIMENTAL SETTINGS

**Datasets.** We perform experiments on five real-world graph datasets with diverse graph data distribution shifts containing: node feature shifts: Cora (Yang et al., 2016) and Amazon-Photo (Shchur et al., 2018); domain shifts (Wu et al., 2020): Twitch-E (Rozemberczki et al., 2021); temporal shifts: Elliptic (Pareja et al., 2020) and OGB-arxiv (Pareja et al., 2020). More details of datasets are listed in Appendix A. For all training, validation, and test graphs, we follow the process procedures and splits in previous works (Wu et al., 2022b; Jin et al., 2023; Wu et al., 2020; Zheng et al., 2023b).

**Test-time Evaluation Protocol.** We test four commonly used GNN models for evaluating GNN generalization under graph distribution shifts following the settings in (Jin et al., 2023), including GCN (Kipf & Welling, 2017), GraphSAGE (Hamilton et al., 2017) (*abbr.* SAGE), GAT (Veličković et al., 2017), and GPR-GNN (Chien et al., 2020) (*abbr.* GPR). For each model, we train it on training

Table 1: Average classification results (%) over the test graphs under various graph distribution shifts on different backbone GNN models. The best results are in bold, and the second-bests are with underlines. 'Rank' indicates the average rank of each algorithm for each backbone; 'OOM' indicates an out-of-memory error on 32 GB GPU memory; TENT with '-' means it cannot be applied to GNNs without batch normalization layers.

| Backbones | Categories | Methods | Amz-Photo | Cora | Elliptic | OGB-Arxiv | Twitch-E | Rank |
|---|---|---|---|---|---|---|---|---|
| GCN | Model-centric | ERM | $88.60_{\pm0.90}$ | $87.49_{\pm7.97}$ | $51.09_{\pm5.63}$ | $38.39_{\pm2.92}$ | $59.80_{\pm3.77}$ | 3.8 |
| | | EERM | $81.05_{\pm0.95}$ | $66.80_{\pm6.51}$ | $45.60_{\pm1.22}$ | OOM | $53.28_{\pm1.88}$ | 6 |
| | | DropEdge | $81.73_{\pm1.23}$ | $74.05_{\pm8.00}$ | $53.83_{\pm4.52}$ | $\mathbf{40.82_{\pm2.18}}$ | $59.49_{\pm4.14}$ | 3.8 |
| | | TENT | $88.60_{\pm0.90}$ | $87.51_{\pm8.01}$ | $47.05_{\pm2.01}$ | $38.45_{\pm2.35}$ | $59.79_{\pm3.77}$ | 3.8 |
| | Data-centric | GTRANS | $\mathbf{89.27_{\pm0.37}}$ | $\underline{95.20_{\pm0.87}}$ | $\underline{56.69_{\pm6.74}}$ | $40.00_{\pm2.30}$ | $\underline{60.38_{\pm3.86}}$ | 1.8 |
| | | TT-GREB (Ours) | $\underline{89.11_{\pm0.47}}$ | $\mathbf{96.12_{\pm1.10}}$ | $\mathbf{57.20_{\pm8.19}}$ | $39.49_{\pm1.72}$ | $\mathbf{60.85_{\pm4.17}}$ | **1.6** |
| SAGE | Model-centric | ERM | $84.03_{\pm7.61}$ | $98.48_{\pm3.68}$ | $57.34_{\pm5.95}$ | $39.26_{\pm2.39}$ | $62.08_{\pm4.04}$ | 4.4 |
| | | EERM | $84.97_{\pm7.26}$ | $96.73_{\pm6.77}$ | $60.94_{\pm5.18}$ | OOM | $61.70_{\pm4.23}$ | 4.6 |
| | | DropEdge | $80.67_{\pm1.61}$ | $92.53_{\pm7.12}$ | $52.84_{\pm3.92}$ | $37.90_{\pm1.74}$ | $\underline{62.19_{\pm4.16}}$ | 4.8 |
| | | TENT | $84.10_{\pm7.71}$ | $98.58_{\pm3.49}$ | $50.16_{\pm3.89}$ | $\underline{39.59_{\pm1.63}}$ | $62.04_{\pm4.06}$ | 3.8 |
| | Data-centric | GTRANS | $\mathbf{89.63_{\pm5.43}}$ | $\mathbf{99.89_{\pm0.03}}$ | $\underline{62.54_{\pm7.94}}$ | $39.49_{\pm2.34}$ | $62.04_{\pm4.06}$ | 2 |
| | | TT-GREB (Ours) | $\underline{88.49_{\pm4.07}}$ | $\underline{99.66_{\pm0.48}}$ | $\mathbf{66.97_{\pm8.94}}$ | $\mathbf{40.15_{\pm1.65}}$ | $\mathbf{62.43_{\pm4.26}}$ | **1.4** |
| GAT | Model-centric | ERM | $91.20_{\pm2.41}$ | $95.53_{\pm4.98}$ | $65.28_{\pm9.59}$ | $40.47_{\pm2.48}$ | $58.23_{\pm3.45}$ | 3.8 |
| | | EERM | $89.13_{\pm4.06}$ | $87.04_{\pm11.07}$ | $50.40_{\pm3.48}$ | OOM | $\underline{59.51_{\pm3.26}}$ | 4.6 |
| | | DropEdge | $69.52_{\pm6.33}$ | $76.71_{\pm4.60}$ | $64.96_{\pm7.12}$ | $\mathbf{43.91_{\pm1.93}}$ | $58.46_{\pm3.35}$ | 4 |
| | | TENT | $91.40_{\pm2.36}$ | $95.57_{\pm4.96}$ | $56.86_{\pm5.10}$ | $30.36_{\pm1.20}$ | $58.23_{\pm3.45}$ | 4 |
| | Data-centric | GTRANS | $\underline{94.04_{\pm0.73}}$ | $\underline{97.28_{\pm2.92}}$ | $\mathbf{66.85_{\pm9.80}}$ | $\mathbf{41.65_{\pm2.26}}$ | $58.20_{\pm3.49}$ | 2.4 |
| | | TT-GREB (Ours) | $\mathbf{94.34_{\pm0.82}}$ | $\mathbf{98.05_{\pm1.03}}$ | $\underline{66.05_{\pm8.92}}$ | $41.45_{\pm2.00}$ | $\mathbf{58.53_{\pm3.50}}$ | **1.8** |
| GPR | Model-centric | ERM | $\underline{87.04_{\pm2.86}}$ | $87.24_{\pm9.11}$ | $64.79_{\pm7.26}$ | $44.38_{\pm2.97}$ | $59.77_{\pm3.73}$ | 3.4 |
| | | EERM | $85.29_{\pm1.48}$ | $\mathbf{89.50_{\pm7.83}}$ | $64.41_{\pm6.97}$ | OOM | $\mathbf{61.76_{\pm4.06}}$ | 3 |
| | | DropEdge | $74.20_{\pm6.90}$ | $73.29_{\pm10.19}$ | $60.62_{\pm6.06}$ | $43.96_{\pm2.37}$ | $59.89_{\pm3.99}$ | 4.6 |
| | | TENT | - | - | - | - | - | - |
| | Data-centric | GTRANS | $86.94_{\pm2.62}$ | $87.45_{\pm8.91}$ | $\underline{67.65_{\pm10.49}}$ | $\mathbf{45.74_{\pm2.24}}$ | $59.89_{\pm3.61}$ | 2.4 |
| | | TT-GREB (Ours) | $\mathbf{88.55_{\pm1.68}}$ | $\underline{88.54_{\pm8.74}}$ | $\mathbf{71.34_{\pm10.01}}$ | $45.14_{\pm2.41}$ | $\underline{60.00_{\pm3.86}}$ | **1.6** |

sets, until the model achieves the optimal node classification on its validation sets following the standard training process, so that we can obtain the 'well-trained' GNN model that keeps fixed in the whole test-time graph rebirth process. We report the average classification performance, and for all experiments, we report the average results of 10 repeated times with different random seeds.

**Baseline Methods.** We compare the proposed TT-GREB with the following baselines that fall in two groups: *graph model-centric methods*: empirical risk minimization (ERM) for standard training (Wu et al., 2022b), data augmentation technique DropEdge (Rong et al., 2019), Explore-to-Extrapolate Risk Minimization (EERM) (Wu et al., 2022b) customized for node-level graph OOD generalization, and test-time training method TENT (Wang et al., 2020); And the recent SOTA *graph data-centric method*: test-time graph transformation method GTRANS (Jin et al., 2023). More demonstrations of the differences among these baselines are presented in Appendix A.

## 4.2 EXPERIMENTAL RESULTS

In Table 1, we report the average node-level classification results over the test graphs under various graph distribution shifts on different backbone GNN models, along with the average rank of each comparison method for each backbone.

As can be observed, our proposed TT-GREB generally delivers great performance across various graph datasets and models, achieving the highest ranks overall: 1.6, 1.4, 1.8, and 1.6 for GCN, SAGE, GAT, and GPR, respectively. These results could verify the outstanding effectiveness of the proposed TT-GREB for modifying graph data at test time to serve better GNN generalization ability.

Moreover, compared with the recent SOTA graph data-centric method GTRANS, our proposed TT-GREB achieves significant improvements in some cases: for example, our method has 5.5%

average improvement of the classification performance from GTRANS's 67.65% to 71.34% on Elliptic dataset with GPR model, and 7.1% average improvement with SAGE model, respectively.

Besides, we can also observe that some model-centric comparison methods, achieve great performance in some cases, for instance, DropEdge and EERM could deliver excellent performance on OGB-arxiv with GCN and Twitch-e with GAT models, respectively. Nevertheless, DropEdge and EERM can not be applied to the test-time application scenario, since it has to modify the training process of GNNs to achieve better generalization ability. Although TENT is suited for the test-time adaption and training scenario, it can not be directly used for models without batch normalization layers, significantly limiting its usage on GNN models.

In summary, our proposed TT-GREB significantly improves GNN generalization for different graph distribution shifts and GNN models during test time, achieving superior average rankings compared to existing approaches. This success is due to the collaboration between the prototype extractor and environment refiner, which enhances the representations of test graph nodes and edges. Additionally, the incorporation of a dual graph contrastive learning objective, coupled with an effective iterative optimization strategy, further contributes to the method's outstanding performance.

## 4.3 ABLATION STUDY OF TT-GREB

In Table 2, we evaluate the effectiveness of the overall framework of the proposed TT-GREB, from the perspectives of sub-module components and learning strategies, respectively. We observe the effectiveness of the prototype extractor (ProExtractor), and the environment refiner (EnvRefiner), respectively. For learning strategies, we test the effectiveness of with and without the dual graph contrastive learning objective (Contrastive_Obj) as well as the iterative optimization method (Iterative_Opt). Baseline denotes the original test graph classification perfor-

Table 2: Ablation study components of the proposed method. For Idx02, ✓* denotes enabling the complete framework in the test-time graph rebirth process but only using the output $G_{\text{Pro}}$ of the prototype extractor for final inference.

| Ablation Index | Sub-module Components | | Learning Strategies | |
|---|---|---|---|---|
| | ProExtractor | EnvRefiner | Contrastive_Obj | Iterative_Opt |
| Baseline | × | × | × | × |
| Idx00 | ✓ | × | × | × |
| Idx01 | ✓ | ✓ | ✓ | × |
| Idx02 | ✓* | ✓ | ✓ | ✓ |
| **Idx03 (TT-GREB)** | ✓ | ✓ | ✓ | ✓ |

mance directly inferring on the well-trained GNNs without any test-time modification. For Idx00 without the contrastive learning objective, the optimization objective would be degraded to the close distance constraint between the output prototype extractor and the original test graph, which means with the weakest supervision signals to instruct the learning process. For Idx02 with ✓*, it denotes that we use the overall proposed framework to give a test-time graph rebirth, but only access the partial output, *i.e.*, the output $G_{\text{Pro}}$ of the prototype extractor for final test-time inference.

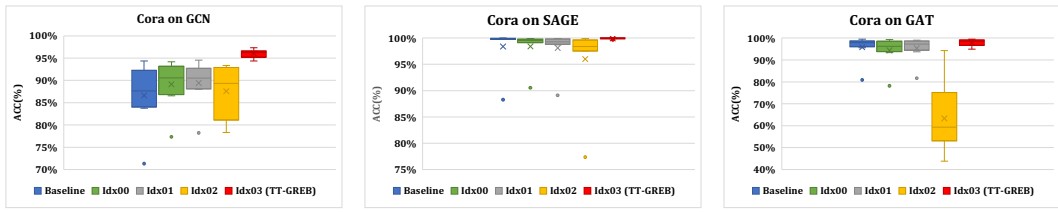

Figure 4: Ablation study results (%) on Cora with GCN, SAGE, and GAT models demonstrated with Box-plot on all test graphs.

The results on all test graphs of Cora on GCN, SAGE, and GAT with a fixed seed run are presented in Fig. 4. As can be observed, our proposed TT-GREB achieves consistently good classification performance on all test graphs on average, also with the smallest standard deviations across all models (shown in Appendix B). Besides, it can also be observed that, generally, each component of sub-modules and learning strategies could contribute to performance improvement in different degrees when these components are coupled together to achieve the best performance of the proposed TT-GREB.

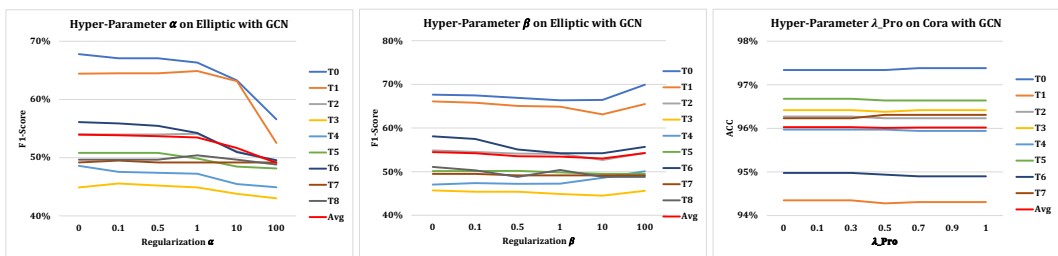

Figure 5: Hyper-parameter sensitivity study results (%) on Cora and Elliptic with GCN model, from left to right: (1) $\alpha$ on Elliptic with GCN, (2) $\beta$ on Elliptic with GCN, (3) $\lambda_{\mathrm{Pro}}$ on Cora with GCN.

## 4.4 HYPER-PARAMETER SENSITIVITY ANALYSIS

In Fig. 5, we evaluate the sensitivity of three hyper-parameters in terms of the regularization in our proposed TT-GREB in Eq. (7) and Eq. (8). Concretely, $\lambda_{\mathrm{Pro}}$ indicates $s = \{1, 2\}$ in Eq. (7) and Eq. (8) for $\lambda_1 = \lambda_2$, corresponding to $\mathbf{W}_{\mathrm{node}}^{\mathrm{Pro}}$, $\mathbf{W}_{\mathrm{edge}}^{\mathrm{Pro}}$, and we empirically set the $\lambda_3 = 1$ for $\mathbf{W}_{\mathrm{node}}^{\mathrm{Env}}$. More observation on the sensitivity of hyper-parameters, *i.e.*, $\lambda_{\mathrm{Env}}$ indicates $\lambda_4$, corresponding to $\mathbf{W}_{\mathrm{edge}}^{\mathrm{Pro}}$ is presented in Appendix B. For regularization weights in balancing the optimization objective, we observe $\alpha$ and $\beta$ in a set of $\{0, 0.1, 0.5, 1, 10, 100\}$. For $\lambda_{\mathrm{Pro}}$ and $\lambda_{\mathrm{Env}}$, we observe the parameter range of $[0, 1]$ with an interval of $0.1$. Observations indicate that the hyper-parameters demonstrate a moderate level of sensitivity within specific ranges, underscoring the robustness of our proposed method to hyper-parameter tuning. More results on visualization are listed in Appendix C.

## 4.5 RUNNING TIME COMPARISON

In Table 3, we compare the running time of our proposed TT-GREB with existing baseline methods, *i.e.*, EERM and GTRANS, which are specifically designed for the graph distribution shift issue. The results are obtained in a single NVIDIA A100 GPU across all datasets with GCN model in 20 epochs. It can be observed that our proposed method achieves a compara-

Table 3: Running time (seconds) comparison in 20 epochs with a single NVIDIA A100 GPU on all graph distribution shift datasets with GCN model.

| Methods | Amz-Photo | Cora | Elliptic | OGB-Arxiv | Twitch-E |
|---|---|---|---|---|---|
| EERM | 14.66 | 2.67 | 230.50 | 191.48 | 22.14 |
| GTRANS | 0.24 | 0.13 | 0.32 | 0.89 | 0.20 |
| **TT-GREB (Ours)** | 2.06 | 1.08 | 1.53 | 4.29 | 1.76 |

ble running time with GTRANS, and significantly exceeds the EERM method, demonstrating its great time efficiency. This efficiency stems primarily from the fact that EERM, a graph data-centric method, necessitates retraining the GNN, which is inherently time-consuming. In contrast, both GTRANS and our proposed method employ test-time graph modifications to enhance performance. However, our method incurs a slight increase in time consumption due to the implementation of a dual iterative optimization strategy.

## 5 CONCLUSION

In this work, we proposed a new graph data-centric method, test-time graph rebirth (TT-GREB), aimed at enhancing the generalization ability of GNN models to test-time graphs affected by distribution shifts through direct manipulation of the test graph data. The overall framework includes a prototype extractor for learning environment-invariant features and an environment refiner for adjusting environment-sensitive features, followed by a dual test-time graph contrastive learning objective and an efficient iterative optimization strategy, facilitating the extraction of optimal prototype and environmental components of the reborn test graph. Our extensive experiments on real-world graph datasets under various test-time distribution shifts confirm the superiority of our method, underscoring its innovative capacity to modify test-time graphs for enhanced GNN generalization. A potential limitation of this work is its current focus on node-level tasks, but future extensions are expected to adapt it for broader applications in graph-level and edge-level tasks.

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

APPENDIX

This is the appendix of our work: **Test-Time Graph Rebirth: Serving GNN Generalization Under Distribution Shifts**. In this appendix, we provide more details of the proposed TT-GREB in terms of more experiments, covering dataset statistics, baseline method comparison, and additional experimental results.

## A    BASELINE METHOD COMPARISON

The statistics of datasets are presented in Table A1. In the following, we demonstrate the differences among these baselines:

- Except for TENT, GTRANS, and our proposed TT-GREB, other baseline methods do NOT perform test-time adaption only with a single-stage training process.

- TENT, GTRANS, and our proposed TT-GREB use two-stage training and test-time adaption, where all the GNN backbones with fixed optimal parameters are trained on common cross-entropy loss under the standard training.

- TENT falls into the model-centric method group by fine-tuning and adapting well-trained GNN models' parameters at the test time, while GTRANS, and our proposed TT-GREB do NOT fine-tune the model parameters but only modify graph data at the test time.

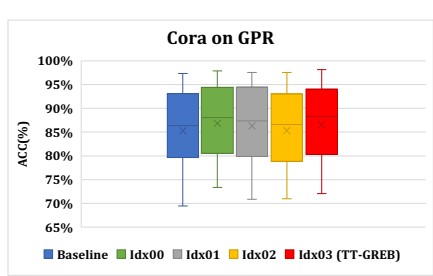

Figure A1: Ablation study results (%) on Cora on GPR model demonstrated with Box-plot on all test graphs.

Figure A2: Hyper-parameter $\lambda_{\text{Env}}$ sensitivity study results (ACC%) on Cora with GCN model.

## B    ADDITIONAL EXPERIMENT RESULTS

We provide more ablation study results covering GPR-GNN model in Fig. A1. Additional hyper-parameter sensitivity analysis results are presented in Fig. A2, where $\lambda_{\text{Env}}$ indicates $\lambda_4$, corresponding to $\mathbf{W}_{\text{edge}}^{\text{Pro}}$ in Eq. (7) and Eq. (8) of the main manuscript.

Note that the outcomes for each hyper-parameter are presented under the condition that the remaining parameters are set to their optimal values. Thus, the optimal set of hyper-parameters is achieved by combining the best values from these analyses.

## C    VISUALIZATION COMPARISON

For a comprehensive understanding of the reborn test graph by our proposed method, in Fig. A3, we present the t-SNE visualization of the original test graph and our reborn test graph on Cora's first test graph, in terms of the output node representations of the well-trained GCN model.

It can be seen that the clusters corresponding to different node class labels are more distinctly separated in the t-SNE latent space after experiencing our proposed test-graph rebirth process. This could effectively verify that the proposed TT-GREB can be beneficial to improve node representation

Table A1: Dataset statistics with various test-time graph data distribution shifts. 'Splits' denotes the number of training/validation/test graphs.

| Distribution shifts | Datasets | #Nodes | #Edges | #Classes | Metrics | Splits |
|---|---|---|---|---|---|---|
| Node feature shifts | Cora (Yang et al., 2016) | 2,703 | 5,278 | 10 | Accuracy | 1/1/8 |
| | Amazon-Photo (Shchur et al., 2018) | 7,650 | 119,081 | 10 | Accuracy | 1/1/8 |
| Domain shifts | Twitch-E (Rozemberczki et al., 2021) | 1,9129,498 | 31,299 - 153,138 | 2 | ROC-AUC | 1/1/5 |
| Temporal shifts | Elliptic (Pareja et al., 2020) | 203,769 | 234,355 | 2 | F1 Score | 5/5/33 |
| | OGB-arxiv (Pareja et al., 2020) | 169,343 | 1,166,243 | 40 | Accuracy | 1/1/3 |

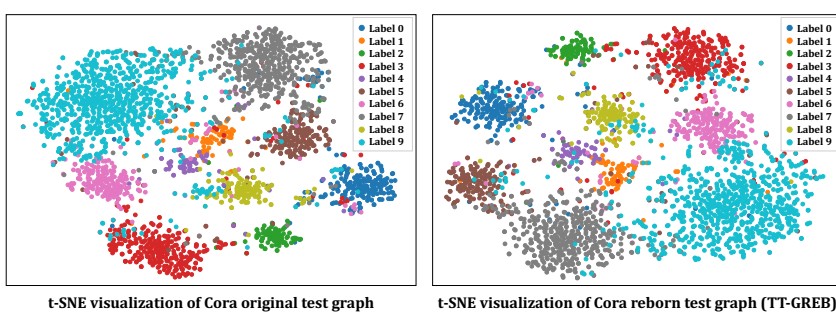

**t-SNE visualization of Cora original test graph**  **t-SNE visualization of Cora reborn test graph (TT-GREB)**

Figure A3: Visualization comparison of t-SNE on the embeddings of the original test graph (1-st test graph) and our reborn test graph with the well-trained GCN model on Cora.

learning, and better separation in the latent space test data demonstrates good generalization ability of our method under graph distribution shifts.

