# OpenReview forum: "Test-Time Graph Rebirth: Serving GNN Generalization Under Distribution Shifts"
_ICLR.cc/2025/Conference — Submitted to ICLR 2025_

### Official Review · Reviewer_AcEo · 2024-10-24

**Soundness:** 3
**Presentation:** 2
**Contribution:** 3
**Rating:** 5
**Confidence:** 4

**Summary:**

In this paper,  the authors introduce a novel graph data-centric paradigm for enhancing the ability of well-trained GNNs to real-world graphs experiencing distribution shifts at test time. The uniqueness of this paper lies in the re-operation of the test data to model the invariant features, which are then fed into the trained GNN for prediction. Empirically, experiments are conducted on five datasets and four benchmark networks to verify the effectiveness of the model.

**Strengths:**

1. The research question is critical
2. It is interesting to design a separate model to capture invariant representations on test data
3. Experimental verification was performed on multiple backbone networks

**Weaknesses:**

1. This method can be seen as the process of designing an unsupervised model to extract invariant features during the test phase. However, it is often difficult to perform decoupled learning under unsupervised conditions.
2. This is misleading in Figure 1(a). In the general graph OOD generalization setting, the test graph is not seen during training. The figure may be a problem setting of domain adaptation.
3. This approach does not seem to be limited to graph data. What are the unique challenges for graph data?
4. In the experiment, the comparison method is not new enough, the latest one comes from ICLR2023. At the same time, Figure 5 is a bit difficult to observe, and its readability can be further improved. If new comparative experiments can be provided here, I will consider improving my score.

**Questions:**

Please see the weaknesses.

---

> ### Author Response · Authors · 2024-11-21
> **Response to Reviewer AcEo (1)**
>
> Thank you for your constructive feedback. We appreciate your recognition of the strengths of our work, particularly the novel graph data-centric paradigm, the uniqueness of the test-data re-operation, and the experimental effectiveness of the model. More detailed responses to your concerns are provided below. We sincerely hope that our detailed responses adequately address your concerns. If our clarifications and evidence resolve your doubts, we kindly ask you to reconsider your evaluation of our work. We value your feedback and are committed to improving the quality and impact of our research.
>
> **【W1: Decoupled Learning under Unsupervision】**
>
> We appreciate the reviewer’s concern regarding the difficulty of decoupled learning under unsupervised conditions during the test phase. This is **precisely the strength of our proposed TT-GREB** — it is designed to address this challenging problem effectively through the following key components:
>
>  - *[Dual-Module Design for Decoupling]* Our proposed TT-GREB utilizes a (1) prototype extractor $f_{\text{Pro}}$ and (b) an environment refiner $f_{\text{Env}}$, which work collaboratively to achieve decoupling. The prototype extractor focuses on isolating stable, environment-invariant features, while the environment refiner adjusts environment-sensitive features to align with the training environment. This targeted division of tasks ensures efficient and accurate decoupling under unsupervised conditions.
>
>  - *[Dual Test-Time Contrastive Learning Objectives with Iterative Optimization]* To enable effective decoupling, we introduce dual contrastive learning objectives. The prototype extractor $f_{\text{Pro}}$ aligns the reborn graph $G_{\text{te}}^{'}$ to ensure that invariant features are retained and emphasized. Simultaneously, the environment refiner  $f_{\text{Env}}$ uses an adversarial contrastive objective to refine environment-specific features to align them with the training distribution. These objectives are coupled with an iterative optimization strategy, enabling progressive refinement and ensuring that both components are effectively optimized during test time.
>
>  - *[Empirical Experimental Results]* The effectiveness of our design is demonstrated in our experimental results. As shown in Table 1, our method achieves superior performance across various datasets under test-time distribution shifts and GNN backbones, reflecting its capability to handle the challenges of unsupervised decoupling during the test phase.
>
> By combining these elements, our method not only addresses the inherent difficulty of unsupervised decoupled learning but also leverages it to achieve robust OOD generalization.
>
> **【W2: Test Graph in Figure 1 (a)】**
>
> We appreciate the reviewer’s observation regarding the test graph in Figure 1(a). The purpose of explicitly depicting the test graph in Figure 1(a) was to **highlight the role of the test graph in the test-time process**, especially for comparing with Figure 1(b), as our method specifically focuses on test-time adaptation of **the test graph**.
>
> We fully agree that in a broader and more general OOD generalization setting, the test graph is not observed during training, and we acknowledge the domain adaptation setting can be taken as a specific OOD generalization scenario for addressing the distribution shift issue. To clarify, Figure 1(a) does not indicate OOD generalization setting but rather emphasizes the test graph's role in test-time adaptation.
>
> Thank you for pointing this out, and we will address it in the updated version of the paper.

---

> ### Author Response · Authors · 2024-11-21
> **Response to Reviewer AcEo (2)**
>
> **【W3: Unique challenges on Graph Data】**
>
> We appreciate the reviewer’s insightful question about the unique challenges of applying this approach to graph data. Our method is specifically designed with graph data in mind. Furthermore, we agree that the proposed conceptual framework and test-time data modifications have the potential to inspire and be adapted for other data types. In particular, we address the following unique challenges associated with graph-structured data, implementing specialized solutions within our approach:
>
> - *[Dependency Between Node and Structure with Complex Distribution Shifts]* Graph data inherently combines node features and graph structures (edges), and node features and graph structure are interdependent. For example, the connectivity of nodes affects feature aggregation in GNNs, while node features influence the significance of edges. This creates a unique challenge where modifications to one aspect (e.g., node features) must be compatible with changes in the other (e.g., edge connections). This interdependence becomes even more challenging when complex distribution shifts occur, such as changes in node feature distributions, alterations in edge connectivity patterns, or simultaneous shifts in both. Our approach addresses this challenge by jointly learning and optimizing both aspects, ensuring a balanced and effective transformation.
>
> - *[Joint Node and Structure Modifications, Learning and Optimization]*
> Due to the interdependent relationship, both node features and graph structures need to be modified harmoniously to achieve effective test-time adaptation. Unlike other data types, where features are typically independent of their structural relationships, graphs require simultaneous updates to node attributes and edge connections to ensure the structural integrity and meaningfulness of the modified graph. Our method explicitly addresses this by optimizing $W_{\text{Node}}^{\text{Pro}}$, $W_{\text{Edge}}^{\text{Pro}}$ for invariant features, $W_{\text{Node}}^{\text{Env}}$, $W_{\text{Edge}}^{\text{Env}}$ for refined environmental features, ensuring coordinated updates.
>
> We conduct experimental validation in graph-specific tasks (e.g., node classification), where it consistently outperforms baselines across different graph datasets and GNN backbones. This empirical evidence further underscores the unique design of our method tailored to graph data.
>
> **【W4: Newer Comparison Method】**
>
> We regret that we could not include newer comparison methods after ICLR2023 in the current experiments. To the best of our knowledge, we have not identified more recent/newer approaches that align with our specific experimental scenario, which focuses on:
> - (1) Unsupervised test-time graph modification;
> - (2) Well-trained and parameter-fixed GNN models;
> - (3) Training-test distribution shifts.
>
> Furthermore, our work addresses a **new research problem within the data-centric** graph modification paradigm, where existing methods are limited. This limitation also highlights **the novelty of our approach**, as it is designed for **practical online application scenarios**, where the deployed GNN model cannot be retrained or fine-tuned. If you could suggest newer and more recent methods that address this specific problem, we would greatly appreciate your recommendations and would be happy to include comparative experiments in this work.

---

> > ### Comment · Reviewer_AcEo · 2024-11-26
> >
> > Thanks for the author's response. After reading other review comments, I tend to maintain the score. Although the author claims that no state-of-the-art methods can be found for comparison, as a data-centric method, can it be used as a plug-in to improve the performance of model-centric methods?  This can improve the practicality of the method. At the same time, the technical introduction and details of the paper seem insufficient for other peers. In summary, in my opinion, the method is interesting, but it may need further improvement in experiments and narratives.

---

### Official Review · Reviewer_oooG · 2024-11-03

**Soundness:** 2
**Presentation:** 2
**Contribution:** 2
**Rating:** 5
**Confidence:** 4

**Summary:**

This paper introduces a data-centric approach called TT-GREB to address the distribution shift in graph learning at test time. Specifically, TT-GREB employs 1. a prototype-extractor module to extract distribution-invariant subgraph, and 2. an environment-refiner module to extract spurious subgraph. The distribution-invariant subgraph, spurious subgraph, and the new graph of the previous two construct a triplet to supervise the graph neural network at test time to mitigate the distribution shift. The prototype-extractor module, environment-refiner module, and the graph neural network are optimized iteratively together. Empirical studies on several datasets show that the proposed TT-GREB can improve the out-of-distribution generalization performance of graph neural network.

**Strengths:**

Improving graph neural networks (GNNs)' out-of-distribution (OOD) performance is important to their real-world deployment since different distribution shifts may occur in the real world. Although some distribution-invariant GNN architectures have been designed, it is still unrealistic to wish these models to generalize to arbitrary unseen distributions. Adapting GNNs at test time is more realistic and general as it can employ some hints from testing data to guide network generalization. This paper follows the test-time adaptation approach and the proposed method can work well on several datasets.

**Weaknesses:**

Yet, the reviewer is concerned about some technical details as follows:

1. Why use two modules to extract the distribution-invariant and spurious subgraphs respectively? According to Proposition 1, the distribution-invariant subgraph and spurious subgraph are fully complementary. Thus, using one module to extract any one of the two subgraphs is sufficient to obtain both distribution-invariant and spurious subgraphs. Instead, this paper introduces two modules to extract these two subgraphs.

2. Based on 1, why is G_{te}^{'} different from G_{te}? As G_{te}^{'} is constructed by the distribution-invariant and spurious subgraphs from G_{te},  G_{te}^{'} should be the same as G_{te} according to 1.

3. As G_{te}^{'} has a spurious subgraph, why push its embedding closer to the embedding of the distribution-invariant subgraph?

4. EERM is designed to improve the OOD generalization ability of GNNs, why does it underperform most baselines in Table 1?

**Questions:**

The authors are encouraged to address the concerns in the Weaknesses part.

---

> ### Author Response · Authors · 2024-11-21
> **Response to Reviewer oooG (1)**
>
> Thank you for your thoughtful review and valuable feedback. We appreciate your recognition of the strengths of our work, particularly the importance to real-world deployment, more realistic and general solutions, and working well on several datasets. More detailed responses to your concerns are provided below. If our responses address your questions and alleviate your concerns, we would be deeply grateful if you could reconsider your evaluation of our work. Your feedback is invaluable to us, and we are committed to further improving our contributions to this new research problem.
>
> **【W1: Two separate environment-varing and -invariant modules】**
>
> We appreciate the reviewer’s question regarding the rationale for using two separate environment-varing and environment-invariant modules. While Proposition 1 highlights that the distribution-invariant subgraph and environmental-varing subgraph are complementary, extracting one from the other in practice is far more complex due to the following reasons:
>
> - [*Nonlinearity of Feature Space*] The mapping and learning of feature spaces in the proposed **(a) prototype extractor for $G_{\text{Pro}}$ and (b) environment refiner for $G_{\text{Env}}$** are inherently nonlinear. While theoretical complementarity exists, it does not guarantee that a single module (e.g., $G_{\text{Env}}$ can effectively disentangle the subgraphs through simple subtraction operations from the original test graph (e.g.,
> $G_{\text{Pro}} \neq G_{\text{te}} - G_{\text{Env}}$). Therefore, we adopt two separate modules to ensure accurate and sufficient extraction of each subgraph.
>
> - [*Distinct Learning Objectives*] Each module has a distinct learning goal tailored to the nature of the respective subgraph. The module for environment-invariant features focuses on retaining stable, predictive patterns consistent across distributions (in Eq.(7)). Meanwhile, the module for environment-varying features requires further refinement (in Eq.(8)) to align the test graph with the training graph environment (in Proposition 1). These fundamentally different objectives necessitate specialized modules for optimal performance.
>
> Empirical evidence in our experiments supports the effectiveness of using two modules. Ablation studies (e.g., Fig. 4) demonstrate that removing or simplifying one module leads to significant performance degradation, highlighting the necessity of this dual-module design.
>
> **【W2: Distinction between $G_{\text{te}}$ and $G_{\text{te}}^{'}$】**
>
> We would like to clarify the *potential misunderstanding* regarding the distinction between $G_{\text{te}}$ and $G_{\text{te}}^{'}$. While $G_{\text{te}}^{'}$ is derived from $G_{\text{te}}$, based on the response on W1, **$G_{\text{te}}^{'}$ is a transformed and enhanced version of $G_{\text{te}}$**, designed to improve OOD generalization through our data-centric approach, **NOT a simple recombination** of two disentangled components. This can be reflected in the following three aspects:
>
> - *[Purposeful Modification on $G_{\text{te}}$ Leads to $G_{\text{te}}^{'}$]* $G_{\text{te}}^{'}$ is specifically constructed to enhance OOD generalization by addressing two critical requirements: (1) extracting stable subgraphs: we extract invariant features $G_{\text{Pro}}$ that capture stable and predictive patterns across distributions (Eq. (7)); (2) refining environmental features: we map the test graph environment-specific features $G_{\text{Env}}$ to align with the training graph environment (Eq. (8) ).
> - *[Fundamental Differences Between $G_{\text{te}}$ and $G_{\text{te}}^{'}$]* $G_{\text{te}}^{'}$ has more stable, model-specific subgraph features and environment-aligned characteristics, which are not present in the original $G_{\text{te}}$. As a result, $G_{\text{te}}^{'}$ is tailored for the parameter-fixed, well-trained GNN model, ensuring improved generalization under distribution shifts during the test time. Thus, $G_{\text{te}}^{'} \neq G_{\text{te}}$.
> - *[Empirical Support from Experiments]* The distinction is further validated by our experimental results. As shown in Table 1, the performance of comparison method "ERM" in Table 1  (using $G_{\text{te}}$) is significantly lower than that of our proposed TT-GREB (using $G_{\text{te}}^{'}$). This demonstrates that extracting stable subgraphs and refining environmental features leads to substantial performance gains, confirming that $G_{\text{te}}^{'}$ is optimized for the task.
>
> We hope this explanation clarifies that $G_{\text{te}}^{'}$ is an enhanced version of $G_{\text{te}}$ during test-time, designed to improve OOD generalization through our data-centric approach.

---

> ### Author Response · Authors · 2024-11-21
> **Response to Reviewer oooG (2)**
>
> **【W3: Embedding Learning for $G_{\text{te}}^{'}$】**
>
> We would like to clarify a potential misunderstanding regarding the reviewer's mention *"spurious".* The graph $G_{\text{te}}^{'}$ does **NOT contain a "spurious" subgraph** in the sense of being inaccurate or irrelevant. Instead, **it consists of** a stable, environment-invariant subgraph and a **"environment-refined" subgraph.** Hence, for its embedding learning, we have:
>  - *[Pushing the Stable Subgraph Closer (Eq. (7))]* In Eq. (7), we align the embedding of the reborn test graph $G_{\text{te}}^{'}$ with the embedding of its stable, environment-invariant subgraph $G_{\text{Pro}}$. This ensures that the predictive, invariant features are emphasized and retained, enhancing generalization to unseen distributions.
>
>  - *[Refining and Pushing Environmental Features away (Eq. (8))]* In Eq. (8), the focus is on refining the environmental features of the test graph. Specifically, the refined environmental features $G_{\text{Env}}$ are **pushed away** from the original, unrefined environment features of $G_{\text{te}}$. This step ensures that the environmental subgraph is adjusted to better align with the training graph’s environment, contributing to the OOD generalization.
>
> Thus, the alignment strategies in $G_{\text{te}}^{'}$ are carefully designed for each component: **the stable, environment-invariant subgraph is pushed closer and the refined environmental subgraph is pushed away** from its unrefined counterpart. We hope this explanation clarifies our method and the rationale behind these embedding learning for $G_{\text{te}}^{'}$.
>
> **【W4: EERM Performance in Table 1.】**
>
> We appreciate the reviewer’s observation and would like to provide further clarification. EERM demonstrates strong performance in certain scenarios. For example, it outperforms other baselines on **Twitch-E with GAT, Twitch-E with GPR, and Cora with GPR**. These results indicate that EERM is effective in certain cases, especially when the distribution shifts are less severe or fall within the range that its environment modeling can address.
> However, in scenarios where EERM performs below other baselines, we attribute this to the following limitations:
>
>  - *[Model-Centric Nature & Training-Time Learning of EERM]* EERM is a **model-centric** approach that primarily **operates during the training phase**, aiming to enhance OOD generalization by simulating distribution shifts in the training data. However, the scope of these simulated shifts is inherently limited. As a result, EERM may struggle to adapt to severe or unanticipated test-time distribution shifts, which are common in real-world applications.
>
>  - *[Restricted Coverage of Environment Variables]* EERM relies on its ability to **model environmental variations during training** but this modeling may not adequately cover the diverse and extreme shifts observed in the test-time graphs. Consequently, EERM's performance may decline in cases where the test graph distributions deviate significantly from what was modeled during training.
>
> These limitations highlight the challenges of relying solely on model-centric methods for OOD generalization. In contrast, data-centric approaches like TT-GREB directly address test-time distribution shifts by modifying the test graph data, making them more adaptable and robust in handling severe distribution shifts. We hope this explanation clarifies the contextual strengths and limitations of EERM's performance.

---

### Official Review · Reviewer_hkp5 · 2024-11-03

**Soundness:** 3
**Presentation:** 2
**Contribution:** 3
**Rating:** 5
**Confidence:** 4

**Summary:**

This paper introduces a test-time data-centric approach, Test-Time Graph REBirth (TT-GREB), designed to improve the generalization abilities of pre-trained Graph Neural Networks (GNNs) under graph distribution shifts at test time. It presents a method for regenerating test-time graphs by separating environment-invariant features (which capture the core predictive patterns) from environment-varying features (which address distributional shifts).  TT-GREB leverages dual test-time graph contrastive learning objective with self-supervision signals, along with an effective iterative optimization strategy to obtain expressive prototype features and environmental features. Extensive evaluations demonstrate TT-GREB's superiority over state-of-the-art methods in enhancing GNN performance under test-time distribution shifts.

**Strengths:**

1. The approach seems novel to me. This paper presents an approach to test-time graph rebirth, which contrasts with existing methods by focusing on modifying architectures and training strategies to address distribution shifts. The motivation is also clear to me.
2. The idea to extract environment-invariant features and refine the environment-varying features is reasonable to me.
3. Experiments include various GNN backbones and baselines to compare the performance.

**Weaknesses:**

1. The experimental results are comparable to the data-centric baseline GTRANS. The improvement is not very evident.
2. The proposed method appears less effective on large-scale datasets, such as the arXiv dataset. For even larger-scale datasets, will the performance be even worse and will the iterative optimization strategy be computationally intensive when the graph sizes are very large?
3. The captions and ticks in figures are too small for readers to read.
4. Some figures have inconsistent font sizes in their captions (e.g., Figure 4), detracting from the professional presentation.

**Questions:**

1. Can the method extend to other model architectures like Graph Transformers?
2. In terms of runtime comparison, could the authors provide more detailed comparisons of training and inference times?
3. How well does TT-GREB generalize to larger-scale datasets beyond those used in the paper? Will it hurt the performance or lead to huge time complexity? Could the authors clarify the scalability of the iterative optimization strategy, especially when applied to very large graph datasets?

---

> ### Author Response · Authors · 2024-11-21
> **Response to Reviewer hkp5 (1)**
>
> Thank you for your constructive feedback. We appreciate your recognition of the strengths of our work, particularly the novel test-time data-centric method, reasonable idea, effective iterative optimization strategy, and superiority over state-of-the-art methods. More detailed responses to your concerns are provided below. Additionally, thank you for pointing out the issues with figure readability and inconsistent font sizes (for W3 and W4), and we will make these improvements in the revised manuscript. We sincerely hope that our detailed responses adequately address your concerns. If our clarifications and evidence resolve your doubts, we kindly ask you to reconsider your evaluation of our work. We value your feedback and are committed to improving the quality and impact of our research.
>
> **【W1: Results compared to GTRANS】**
>
> We thank the reviewer for raising this point regarding the result improvement over the baseline method GTRANS. Below, we address and clarify this from two key perspectives: methodological comparison and experimental validation.
>
> - **#[Ours TT-GREB: Tailored Design for Distribution Shifts vs. GTRANS: Fully Parameterized Approach]#** TT-GREB is specifically designed to handle complex graph distribution shifts by incorporating (1) a prototype extractor and (2) an environment refiner. In contrast to GTRANS, which employs a fully parameterized matrix approach, our method TT-GREB effectively decomposes and reassembles the test-time graph to align better with training graph distributions. Considering that both methods adopt a data-centric paradigm, the degree of graph distribution shifts and the impact of environment-varying components play a critical role in experimental outcomes. When the environment-varying factors significantly contribute to distribution shifts, our proposed method TT-GREB demonstrates superior performance than GTRANS.
>
> - #[Experimental Validation]# As shown in Table 1, TT-GREB achieves a notable 5.5% increase in F1 score on the Elliptic dataset with the GPR model (**Ours TT-GREB: 71.34%** vs. GTRANS: 67.65%) and a 7.1% improvement with the SAGE model (**Ours TT-GREB: 66.97%** vs. GTRANS: 62.54%), significantly outperforming GTRANS in scenarios characterized by pronounced distribution shifts.
>
> **【W2&Q3: Clarify Large-scale Dataset Performance and Computation Intensity】**
>
> We thank reviewer's insightful observation about test-time graph rebirth problem on large(r)-scale datasets like arXiv. Below, we address the concerns about scalability and computational intensity:
> - [Node-Level Task and Data-Centric Methods] As typical data-centric methods, both GTRANS and our TT-GREB require modifications to the **entire graph** during test time, as working on the **node-level** graph learning task. Consequently, as graph sizes increase (e.g., with more nodes and edges), this approach leads to heightened computational demands. For example, as noted in Table 1, the comparison method *EERM even encounters out-of-memory (OOM)* errors on the arXiv dataset, underscoring that handling large-scale datasets remains a significant future challenge for test-time data-centric methods. Recognizing this, we are eager to explore strategies in future work to alleviate such scalability limitation for test-time graph rebirth.
> - [Computational Intensity of Iterative Optimization]
> Due to above mentioned data-centric property, we acknowledge that the computational complexity scales with graph size. We provide more detailed analysis in terms of computational intensity of our proposed iterative optimization strategy as follows:
>
> (1) Reweighting Edges: Operations over ${\boldsymbol{W}}_{\text{edge}}$ $\in \mathbb{R}^{N \times N}$ scale with $O(N^{2})$. For graphs with a very large number of nodes $N$, this can still become computationally intensive.
>
> (2) Iterative Nature: The iterative updates add a scaling factor $T$ (number of iterations) to the complexity, where each iteration involves:
> - (2.1) Forward and backward passes through the GNN, scaling with $O(N F d+N F^{2}+N^{2}F)$
> - (2.2) Gradient updates and regularization for reweighting matrices $O(N^{2}+d^{2})$
>
> These computations allow our proposed TT-GREB feasible for moderately large graphs. As shown in Table 3, the running time remains manageable (e.g., 4.29 seconds for OGB-arXiv on a single NVIDIA A100 GPU for 20 epochs). Nevertheless, *computational demands observed for larger graphs are influenced by the data characteristics of node-level task*. We will put these detailed analyses in the revised Appendix. We sincerely appreciate the reviewer’s feedback.

---

> ### Author Response · Authors · 2024-11-21
> **Response to Reviewer hkp5 (2)**
>
> **【Q1: Extension to Graph Transformers】**
>
> Thank you for the insightful question. Yes, the proposed method can be extended to other model architectures like Graph Transformers.
> Our proposed method is data-centric and model-agnostic, mainly working on operating graph data at test time, which does not depend on the specific architecture of the underlying graph model. For Graph Transformers, the reweighting process can be directly applied to the input features and attention mechanisms, ensuring compatibility with their architecture.
> Extending TT-GREB to Graph Transformers would involve adapting the reweighting matrices to align with the attention-based computation of Graph Transformers, which we believe is a natural extension.
>
> **【Q2: More Detailed Comparisons of Training and Inference Times】**
>
> Thank you for your question regarding more detailed runtime comparisons of training and inference times. In **Table 3, the runtime provided refers to the Test-time Training Time (s)**. For convenience, we have included it here for easier comparison. Additionally, below we provide a detailed Test-time Inference Time (seconds) for TT-GREB compared to GTRANS across various datasets, with a single NVIDIA A100 GPU:
> | Test-time Training Time (s) from Table 3   | Amz-Photo | Cora   | Elliptic | OGB-Arxiv | Twitch-E |
> |--------------------------------|-----------|--------|----------|-----------|----------|
> | GTRANS                         | 0.24    | 0.13 | 0.32   | 0.89    | 0.20  |
> | TT-GREB (Ours)                 | 2.06    | 1.08 | 1.53   | 4.29    | 1.76   |
>
> | Test-time Inference Time (s)   | Amz-Photo | Cora   | Elliptic | OGB-Arxiv | Twitch-E |
> |--------------------------------|-----------|--------|----------|-----------|----------|
> | GTRANS                         | 0.0003    | 0.0002 | 0.0022   | 0.0024    | 0.0025   |
> | TT-GREB (Ours)                 | 0.0294    | 0.0143 | 0.0239   | 0.0504    | 0.0477   |
>
> While TT-GREB incurs a slightly higher runtime compared to GTRANS, the difference remains manageable. This additional cost is justified by the improved performance it achieves in handling test-time distribution shifts. We are committed to exploring further optimizations to reduce runtime overhead while preserving the advantages of our approach. Thank you for highlighting this important aspect, and we will add this to the revised Appendix.

---

### Official Review · Reviewer_mSYU · 2024-11-04

**Soundness:** 2
**Presentation:** 2
**Contribution:** 2
**Rating:** 3
**Confidence:** 3

**Summary:**

The paper presents a novel data-centric approach called Test-Time Graph Rebirth (TT-GREB) to improve the generalization of GNNs under distribution shifts. The method manipulates test graph data directly and does not require modifications to the GNN architecture or retraining. To achieve this, the method decomposes the test graph into environment-invariant and environment-varying components, using a prototype extractor and environment refiner. This leads to the creation of a “reborn” test graph that better matches the training graph distribution, enhancing the GNN’s generalization without altering its original structure. The dual contrastive learning objective and iterative optimization are key to refining the test graph for improved node classification performance.

**Strengths:**

The paper introduces a practical and innovative solution to the problem of GNN generalization under test-time distribution shifts without requiring retraining or modifications to the GNN model itself. The dual contrastive learning framework is designed to refine test-time graphs by optimizing environment-invariant and environment-varying features. Experimental results on diverse real-world datasets demonstrate that TT-GREB outperforms existing methods.

**Weaknesses:**

1. The problem setup seems quite contrived. Particularly, it is mostly unclear why a graph could be explicitly decomposed into environment-invariant and environment-varying parts, where the environment-varying variables are upper bounded with a certain notion of discrepancy (which is also not formally defined). While the authors write this as a proposition, it seems more like an (unrealistic) assumption without any support or proof.
2. There is no guarantee that the proposed Test-time Graph Rebirth as in definition 3.1 can improve out-of-distribution generalization.
3. The method only shows marginal improvement over another data-centric graph ood method.

**Questions:**

See above

---

> ### Author Response · Authors · 2024-11-21
> **Response to Reviewer mSYU**
>
> Thank you for your thoughtful review and valuable feedback. We appreciate your recognition of the strengths of our work, particularly the novelty of our test-time graph rebirth (TT-GREB) framework, its practical design, and its outperforming performance. More detailed responses to your concerns are provided below. If our responses address your questions and alleviate your concerns, we would be deeply grateful if you could reconsider your evaluation of our work. Your feedback is invaluable to us, and we are committed to further improving our contributions to this new research problem.
>
> 【**W1-1: Problem setup for environment-varying and environment-invariant decomposition**】
>
> #[For environment-varying part]# Potential unknown environmental variables causing distribution shifts between training and test data is a common problem setup in addressing the Out-of-Distribution (OOD) generalization problem. For example, one of our baseline methods, Explore-to-Extrapolate Risk Minimization (EERM) (Wu et al., 2022b), adopts such problem setup.
>
> #[For environment-invariant part]# Learning invariant representations can be a key approach to address such distribution shift problem, also highlighted by (EERM method). We provide evidence supporting the problem setup for further clarification, as cited below:
> > - *[environment-varying] "Recent studies of the OOD generalization problem like Rojas-Carulla et al. (2018); Bühlmann (2018);Gong et al. (2016); Arjovsky et al. (2019) treat the cause of distribution shifts between training and testing data as a potential unknown environmental variable e."*
> > - *[environment-invariant part] "Several up-to-date studies develop new objective designs and algorithms for learning invariant models, showing promising power for tackling OOD generalization (Chang et al., 2020; Ahuja et al., 2020;Krueger et al., 2021; Liu et al., 2021; Creager et al., 2021; Koyama & Yamaguchi, 2021)."*
>
> Additionally, other related works on environment-invariant and environment-varying representation learning, as mentioned in Lines 041–043 of our manuscript, include:
> > - “The main reason behind such distribution mismatch lies in the underlying environmental variations, with time-related attribute changes, agnostic corruptions, and inconsistent graph data collection procedures (Sui et al., 2023; Chen et al., 2023b; Jin et al., 2023).”
>
> 【**W1-2: Assumption on environment-varying variables with bounded discrepancy**】
>
> The assumption that "A = environment-varying variables are bounded by a certain notion of discrepancy" is a sufficient but not necessary condition for "B = the GNN model $GNN_{{\theta}^{*}}$
> that has been well trained on the $G_{tr}$ would keep good generalization on the test graph with $A \rightarrow B$.
> - *Sufficiency:* Bounding the discrepancy $dist(\cdot)< \epsilon$ simplifies the problem and aligns with many practical scenarios, such as cases where environmental variations are minor perturbations or exhibit predictable patterns. In the ideal case, if there is no discrepancy between $G_{tr}^{Env}$ and $G_{te}^{Env}$, the distributional shift due to environment-varying components would be minimal, allowing the GNN model $GNN_{{\theta}^{*}}$ to naturally generalize well on $G_{te}^{Env}$.
> - *Non-necessity:* The GNN may still generalize well even if this assumption is not met. For example, if the environment-varying variables have limited influence on the model’s predictions or if the model primarily relies on environment-invariant features $G^{Inv}$, the generalization error may remain low even without bounding the discrepancy.
>
> **In the following, we provide more detailed proof.**
>
> (1) Training Objective: The GNN is trained to minimize a loss $L(G_{tr}; \theta_{tr})$, which can be decomposed into:$L(G_{tr}; \theta_{tr}) = L^{Inv}(G_{tr}^{Inv}; \theta_{tr}) + L^{Env}(G_{tr}^{Env}; \theta_{tr})$. Here:
> $L^{Inv}$ is influenced by the invariant component, and since $G_{tr}^{Inv} \sim Q^{Inv}$, this term generalizes well.
> $L^{Env}$ captures the environment-varying part, with a discrepancy bounded by $\epsilon$, as defined.
>
> (2) Generalization Bound: The generalization error between $G_{tr}$ and $G_{te}$ can be expressed as:
> $|L(G_{tr}; \theta_{tr}) - L(G_{te};\theta_{tr})| \leq |L^{Inv}(G_{tr}^{Inv};\theta_{tr}) - L^{Inv}(G_{te}^{Inv}; \theta_{tr})| + |L^{Env}(G_{tr}^{Env}; \theta_{tr}) - L^{Env}(G_{te}^{Env}; \theta_{tr})|$.
>
> - The first term is 0, as $G_{tr}^{Inv} \sim Q^{Inv}, G_{te}^{Inv} \sim Q^{Inv}$.
>
> - The second term is bounded by $\epsilon$, which dominates the total generalization error, ensuring that $GNN_{{\theta}^{*}}$ can generalize well to $G_{te}$.
>
> Thanks for your suggestion and we will provide this more detailed proof in the appendix of our revised manuscript.

---

> ### Author Response · Authors · 2024-11-21
> **Response to Reviewer mSYU (2)**
>
> **【W2-NO guarantee in definition 3.1.】**
>
> Definition 3.1 introduces "test-time graph rebirth" **as** a data-centric **problem definition**, **NOT as a theoretical proposal to claim guarantee** for OOD generalization. Its primary purpose is to define the transformation of the original test graph $G_{te}$  to a reborn graph $G_{te}^{'}$, with a learnable function $g(\cdot)$. Within this proposed **new research problem** (which is one of our core contributions), any future method aimed at designing the learnable function $g(\cdot)$ to achieve this goal can be regarded as a test-time graph rebirth approach.
>
> For how the proposed method guarantees OOD generalization improvement, empirically, it lies in the proposed dual test-time graph contrastive learning objective, building on the contrastive learning paradigm introduced by (Jin et al., 2023). Theoretical support for this can be found in the proof provided in the response to W1-2.
>
> **【W3-Improvement over another data-centric graph OOD method】**
>
> We appreciate the reviewer pointing out this question. Our proposed TT-GREB demonstrates consistent and significant improvements **across diverse datasets and GNN backbones**, achieving the highest average ranks in all scenarios (e.g., **1.6 for GCN, 1.4 for SAGE, 1.8 for GAT, and 1.6 for GPR**, as shown in Table 1. For example, TT-GREB improves the classification performance on **Elliptic by 5.5% with GPR and 7.1% with SAGE**, compared to GTRANS, clearly showcasing its effectiveness in addressing severe distribution shifts.
>
> In addition to its quantitative improvements, TT-GREB contributes (1) a new research problem: test-time graph rebirth; (2) innovative dual contrastive learning solution with iterative optimization. All these provide enhanced robustness under distribution shifts, as demonstrated in the ablation studies (Fig. 4). Furthermore, TT-GREB maintains competitive computational efficiency, as shown in Table 3, while delivering these benefits without requiring model retraining or fine-tuning.

---

> ### Author Response · Authors · 2024-12-03
> **Inviting Discussions with Reviewer mSYU**
>
> Dear Reviewer mSYU,
>
> We appreciate your thoughtful feedback and understand that your concerns may pertain to the theoretical proof and the problem setting. We have addressed these points in our response and kindly ask if you could **revisit them to see if they alleviate your concerns and prompt a reconsideration of your ratings for our work**. Your suggestions are invaluable to us.
>
> "Test-Time Graph Rebirth" introduces a **new research problem**, aiming to bridge this practical scenario with the research community. We greatly value your insights and support in fostering this initiative.
>
> Best regards,
>
> The "Test-Time Graph Rebirth" Authors

---

### Author Response · Authors · 2024-11-21
**Common Response to All Reviewers**

We sincerely thank all the reviewers for their constructive feedback, insightful comments, and recognition of the strengths of our work. We are particularly grateful for the recognition of:

- *Important/critical research problem* of our proposed Test-time Graph REBirth (TT-GREB) to real-world deployment (**Reviewer oooG, AcEo**)
-  *Novel/practical and innovative/uniqueness/reasonable/interesting/more realistic and general/ approach* of graph data-centric paradigm for test-time data modification, under test-time distribution shifts (**All Reviewers!**)
-  *Effective iterative optimization strategy* with optimized *environment-invariant and environment-varying features* (**Reviewer hkp5, Reviewer mSYU**)
- *Outperforming* experimental results on *multiple datasets and GNN backbones* over state-of-the-art methods  (**All Reviewers!**)
- Clear motivation (**Reviewer hkp5**)

We are grateful for your positive recognition and have carefully addressed your insightful concerns in the detailed responses below. If our explanations and additional evidence resolve your questions, we would greatly appreciate it if you could reconsider your evaluation of our work. Thank you again for your thoughtful reviews and valuable feedback.

---

### Meta-Review · Area_Chair_fWSn · 2024-12-22

**Metareview:**

(a) The paper proposes a method called Test-Time Graph REBirth (TT-GREB) to improve the generalization of graph neural networks (GNNs) under distribution shifts. It aims to address the issue by directly manipulating test graph data through a prototype extractor and an environment refiner. The key findings are that TT-GREB can outperform existing methods in enhancing GNN performance on diverse real-world datasets under test-time distribution shifts.

(b) The strengths of the paper include proposing a new data-centric approach for test-time graph modification, introducing a dual contrastive learning framework for optimizing test-time graphs, and demonstrating effectiveness through extensive experiments with various GNN backbones and datasets.

(c) The weaknesses are that the problem setup seems contrived with unclear assumptions (e.g., graph decomposition), the improvement over a data-centric baseline is not highly significant, and the method may face scalability issues on large-scale datasets with high computational intensity.

(d) The main reasons for rejection are the lack of clear justification for the proposed approach, especially regarding the assumptions. Although the experimental results show some improvements, they are not overwhelmingly convincing, and the concerns about scalability and the overall problem formulation remain unresolved.

**Additional Comments On Reviewer Discussion:**

Reviewers raised concerns about the problem setup (e.g., graph decomposition and assumptions), the marginal improvement over baselines, the effectiveness on large-scale datasets, and the clarity of presentation (e.g., figure readability). Authors addressed these by providing justifications for the problem setup with references, clarifying the differences between the proposed method and baselines in terms of design and experimental results, analyzing the scalability and computational complexity on large-scale datasets, and promising to improve the presentation. However, in the final decision, the concerns about the clear justification for the proposed approach and the overall significance of the improvement were still considered major drawbacks, outweighing the authors' responses.

---

### Decision · Program_Chairs · 2025-01-22

Reject